# Video Game Playing and Internet Gaming Disorder: A Profile of Young Adolescents

**DOI:** 10.3390/ijerph20247155

**Published:** 2023-12-08

**Authors:** Marta Labrador, Iván Sánchez-Iglesias, Mónica Bernaldo-de-Quirós, Francisco J. Estupiñá, Ignacio Fernandez-Arias, Marina Vallejo-Achón, Francisco J. Labrador

**Affiliations:** 1Department of Personality, Assessment and Clinical Psychology, Complutense University of Madrid, 28223 Madrid, Spain; marlabra@ucm.es (M.L.); mbquiros@psi.ucm.es (M.B.-d.-Q.); fjepuig@ucm.es (F.J.E.); igfernan@ucm.es (I.F.-A.); mvalle02@ucm.es (M.V.-A.); flabrado@psi.ucm.es (F.J.L.); 2Department of Psychobiology & Behavioral Sciences Methods, Complutense University of Madrid, 28223 Madrid, Spain

**Keywords:** video games, children, adolescents, addictive behavior, internet, video gaming disorder

## Abstract

In recent times, growing concern has arisen regarding the utilization of technology, video games, and the emergence of internet gaming disorder (IGD), particularly among young adolescents. This worry arises from the ambiguity in distinguishing between “normal” and “problematic” video game behavior, despite efforts to establish clear criteria for defining both. The goal of this study is to outline distinct profiles of adolescent video game players and identify variables associated with their gaming practices that correlate with problematic gaming. The study utilizes a substantial sample of adolescents drawn from a representative cross-section of educational institutions in the city of Madrid, ranging in age from 12 to 16 years. In total, 1516 participants (75%) acknowledged engaging in video game activities. The research delves into characterizing prevailing profiles of video game participants within this cohort and scrutinizes the profile that aligns with issues of IGD. In summary, approximately three-quarters of young adolescents participate in video gaming, with males constituting the majority. Typically, participants immerse themselves in action genre games for over three days per week, with males exhibiting a higher frequency than their female counterparts. Elevated gaming frequency correlates with heightened IGD scores, particularly among females. Young adolescents show a preference for game consoles (males) and mobile phones (females) and often play alone at home. Specific factors such as the device used, online mode, company, and gaming location impact the IGD scores. These profiles aim to assist families and educators in recognizing potential risk behaviors and IGD concerns; however, it is crucial to emphasize the necessity for case-specific screening and evaluation before deliberating on such behaviors.

## 1. Introduction

While historically there has been minimal concern about behavioral addictions in young adolescents, there is currently substantial social alarm regarding the use of video game playing (VGP) [1], and particularly online gaming. The Diagnostic and Statistical Manual of Mental Disorders, DSM-5 [1], defined internet gaming disorder (IGD) as “a pattern of excessive and prolonged Internet gaming that results in a cluster of cognitive and behavioral symptoms, including progressive loss of control over gaming, tolerance, and withdrawal symptoms, analogous to the symptoms of substance use disorders”. The DSM-5 [1] tried to shed light on this issue by including IGD among the “categories under study”, highlighting nine diagnostic criteria. At least five must be met to establish a diagnosis of IGD. For several reasons, this contribution, although important, does not overcome the problem of distinguishing “normal” from “abnormal”. One such reason is the lack of precision in some criteria, such as criterion A “Persistent or repeated use of the Internet to play games…” and criterion 6 “Continuous excessive use of games…” [1]. What criterion tells us whether the VGP is persistent, excessive, or continuous? In addition, these nine criteria do not profile IGD equally. Thus, Király et al. [2] found that the criteria of “continuation”, “concern”, “negative consequences”, and “escape” were associated with less severe IGD, while “tolerance”, “loss of control”, “abandonment of other activities”, and “disappointment” were associated with more serious levels of IGD. Finally, “worry” and “escape” provided little information. Alternatively, Rehbein et al. [3] found that the symptoms relating to “abandonment of other activities”, “tolerance”, and “withdrawal” were the most relevant to the diagnosis of IGD. The value of distinguishing online and offline VGP behavior has also been questioned [4]. In addition, seven of the nine criteria for IGD are identical or derived from those for Pathological Gaming, which questions their specificity and precision. Finally, the consideration of IGD as an addiction and its diagnosis have caused controversy [5,6,7,8].

This imprecision in the boundaries of the problem (IGD, addiction, or problematic VGP) makes it easier for studies aiming to identify its magnitude to produce very different percentages of IGD prevalence: 1.2% of German students [3]; 1.6% (5.1% at risk of IGD) in a sample of young adolescents across five European countries [9]; 5.7% when examining IGD among Dutch students [10]; 34.0% in Taiwan [11]; and 5.0% in Spain [12]. A meta-analysis [13] estimates that the prevalence of IGD is between 0.7% and 15.6%. In addition to the imprecision of the DSM-5 criteria, this difference in results comes from the possible range of diagnoses (IGD, addiction to VG, or problematic VGP) and the use of different assessment tools. Reviews of VGP assessment tools detected significant problems, concluding that the instruments used are generally inconsistent [14,15,16].

The number of hours they play, something that is easy to verify, along with the level of attention absorbed, stand out, as observed in the development of other addictions, where time spent on smartphones emerges as the most influential predictor of social media addiction [17]. Young adolescents have a fundamental need for connection and a sense of belonging to a group, and they resort to the internet to establish connections. Therefore, this resource is an effective means to achieve their objective [17,18]. However, there are other aspects, such as whether the VGP affects their school work, replaces other activities, or restricts their social relationships, which may be also relevant in distinguishing “normal” from “problematic” behavior. Attempts have been made to establish the criteria for making this distinction. Thus, equating “problematic” VGP to addiction highlights six dimensions to identify it: salience, mood alteration, tolerance, withdrawal, conflict, and relapse [19]. Subsequently, Demetrovics et al. [20] revised these dimensions as follows: preoccupation, excessive use, immersion, social isolation, interpersonal conflicts, and withdrawal.

Despite the problems in specifying when VGP behavior is “normal”, some figures do stand out:(a)A high percentage of young adolescents, more males than females, play VG. Most European adolescents played VG regularly (60.5%; 84.7% male and 42.8% female) [9]. Similar results were found in 13–15-year-old Dutch children (60.0%; male and 13.5% female) [21]. Another study found that 85.5% had played VG at some point [22], and another one found that 55.6% of adolescents had played a VG in the previous month, (65.4% male) [23].(b)The time dedicated to VGP seems high. Merelle et al., looking at Dutch children (12–15 years old), found an average VGP time of 15 h per week for males and 7 h for females [10]. Király et al., with Hungarian adolescents (average age of 16.4 years old), found that 11.2% played VG online for more than 7 h a day, 20.6% for more than 5 h, and 68.3% for more than 2 h [22]. Pontes et al. looked at people aged 16–58 and found that 26.0% played VG for more than 30 h per week [24].(c)Correlations appeared between the time spent VGP and gaming problems [9,22]. Players with a high risk of developing IGD are mainly male, play for more than 5 h a day, and have lower school grades, lower self-esteem, and more depressive symptoms [23].(d)Negative consequences are associated with an increase in time spent VGP. Psychosocial and psychopathological problems stand out, along with a reduction in academic performance [9,10,25,26,27].(e)A special relationship between VGP problems and certain types of games exist, especially massively multiplayer online role-playing games (MMO) and shooting games [9].

Although these figures can provide some information about VGP among young adolescents, there are large variations, and the landscape is not stable. Therefore, figures from the previous five years may not be very descriptive of the current situation.

Given these considerations, it seemed important to identify the current normal behavior of young adolescents when playing VG, both online and offline. This would allow us to establish what is the standard “norm” or “average” for VGP, which can then be used as a benchmark against which to compare the VGP behavior of any young adolescents. Given these considerations, this study was designed to profile the VGP behavior of young adolescents of 12 to 16 years of age attending school in Madrid. Furthermore, we studied the relationship between some characteristics of this profile and IGD.

## 2. Materials and Methods

### 2.1. Participants

Using stratified random sampling of the schools in Madrid, 2020 students were selected from 37 different schools (47.5% female), aged 12–16 years old (M_age_ = 13.8, SD_age_ = 1.3) and in different school years. Of the participants, 41 (2.0%) worked in addition to studying. Of the initial sample, 1516 reported to play VG (32.4% female, M_age_ = 13.8, SD_age_ = 1.4). Table 1 shows the educational level of VG players.

### 2.2. Instruments and Variables

#### 2.2.1. Gamertest

This is an expert online system designed to detect the problematic use of VG, developed by our research group, which consists of eight sections: (1) demographic data; (2) video game habits; (3) level of risk of problems with VGP (including the IGDS9-SF described below); (4) engagement in the game; (5) attitudes toward video games; (6) cognition on video games; (7) level of self-control/impulsiveness; and (8) emotional unease. For this study, we have only used items (1) (gender, age, and school year), (2), and (3); the psychometric properties of the measurement instrument are detailed in the following section. The complete instrument can be located at the following website: http://www.famgi14.es/gamertest/index.html (accessed on 1 December 2023).

#### 2.2.2. Internet Gaming Disorder Scale—Short-Form, IGDS9-SF

The IGDS9-SF is a validated Spanish translation [28] of the original English scale [29], which is aligned with the criteria for IGD as outlined in the DSM-5 [1]. The scale comprises nine 5-point Likert items, ranging from 1 “Never” to 5 “Very Often.” It is designed to gauge the severity of IGD and the adverse effects of online and offline video gaming over the past 12 months. In various subgroups, it demonstrated acceptable internal consistency (with ω values ranging from 0.778 to 0.828) and split-half reliability (with *R*_xx_ values ranging from 0.770 to 0.822). The scale exhibits a single-factor structure. The total score is calculated as the sum of item scores, where a higher score indicates greater IGD severity. This scale is useful for the measurement of IGD in participants from 12 to 22 years old, and offers a scoring scale for measurement purposes, segmented by gender.

### 2.3. Procedure

The 37 schools that participated in this study were chosen randomly from a list of schools in the city of Madrid [30], segmented by each of the 21 city districts and type (public, private, or state-subsidized schools).

The first version of Gamertest, developed from the literature review, was reviewed by six expert judges in gaming and VGP. Incorporating their contributions, a pilot study was carried out with 20 young adolescents (snowball sampling), taking Gamertest and providing suggestions. The relevant suggestions having been incorporated, and five independent evaluators, graduates in psychology, were trained in its application. A second pilot study was conducted with pupils from a school who obtained their parents’ consent, incorporating the relevant changes. Finally, Gamertest was taken by the pupils from the 37 selected schools who obtained their guardians’ consent. The test, anonymous and grouped, was carried out on computers in the computer rooms at the schools, the average time needed to complete it being estimated at 30–40 min. The responses of the participants were collected and coded directly into the computer database.

### 2.4. Data Analysis

We analyzed the data of the 1516 VG players. We used frequency tables for categorical variables. For quantitative variables, we computed the mean and standard deviation (plus the minimum, maximum, median and interquartile range, IQR, in non-normal distributions). As bivariate analyses, Pearson’s correlation coefficient (Spearman correlation for non-normal variables) was calculated, along with the X2 for categorical variables. To study the differences between the two groups, a t-test and ANOVA were used. To specify the size of the effect, r^2^, η^2^, and the contingency coefficient C were calculated

A statistical power of 1—β = 0.869—was calculated in a one-way ANOVA test, comparing players (*n* = 1516) among five groups (as in the “preferred place for video gaming” comparison), setting α at 0.050, even assuming a small effect size *f* = 0.10. This analysis was performed using G*Power 3.1.9.2 (Heinrich-Heine-Universität Düsseldorf, Düsseldorf, Germany) [31], and the rest using SPSS 25.

## 3. Results

In total, 1516 participants (75.0%) stated that they played VG, with a greater proportion for males (96.6%) than females (51.2%), X^2^ (1) = 554.65, *p* ˂ 0.001, C = 0.464. There were no differences in average age between players and non-players, t(897.24) = −1.73, *p* = 0.084. The IGD scores were significantly higher for male participants (M = 17.41, SD = 6.36) than female ones (M = 13.52, SD = 5.17), t(1411.15) = 14.82, *p* < 0.001, r^2^ = 0.078. This was to be expected, due to the differences in gender found in the validation process of the IGDS9-SF [28]. However, within the sample range, age was not significantly related to the IGDS9-SF, r = 0.008, *p* = 0.742.

Table 2 shows how many types of VG they played. The number of different types were positively correlated with the IGD score, for male, r = 0.137, *p* ˂ 0.001, r^2^ = 0.020, as well as for female participants, r = 0.241, *p* ˂ 0.001, r^2^ = 0.058. The most frequent types of VG were “Action and Adventure”, “Sports”, and “Massively Multiplayer Online (MMO)”. Table 3 shows the frequency of use of each type of VG, the proportion of males being significantly higher in 7 of the 13 games.

Table 4 and Table 5 show data on gaming frequency, involvement, and preferences, by gender. Table 4 shows the favorite types of games. Participants could indicate up to three different types as a favorite game. What stands out is the lack of dispersion; two types of game were within the favorites for 58.4% and 42.9% of the players.

Table 5 shows the frequency of gaming in terms of days per week, finding significant relationships between the gender of the participants and the number of days on which they play each week. More hours of playing VG per week were associated with a higher IGDS9-SF score; we found a significant linear relationship for male, F(1, 1017) = 83.62, *p* < 0.001, η^2^ = 0.075, and female participants, F(1, 484) = 103.48, *p* < 0.001, η^2^ = 0.165.

Table 6 shows the preferences of the players depending on the device, the place where they played, online/offline mode, and the company. Table 7 shows the statistics for IGDS9-SF scores by gaming preferences. One-way ANOVA was carried out to assess the differences in the scale scores for each VG modality. We found significant IGDS9-SF score differences in all tests, except for the preferred place in the woman subsample. The difference in IGD for company in the woman subsample stands out. With a 6.4% of variance accounted for, the IGDS9-SF scores ranged from 11.2 (company of one person online) to 19 (company of a group online). On the IGDS9-SF scoring scale [28], a score of 11 equals a percentile score of 45, while a score of 19 is a percentile score between 85 to 90 in IGD. For ease of reference and the interpretation of the results, we have included the scoring scale in Appendix A.

## 4. Discussion

This study characterized video game player profiles from a substantial sample of pupils of both sexes, aged between 12 and 16 years, representing schools in Madrid. We also explored the relationship of these profiles with internet gaming disorder (IGD).

In a larger initial sample, three-quarters of the participants reported playing video games (VG), a percentage higher than the 60.5% and 60% found in other studies [9,21]. This difference may be attributed, among other factors, to the increasing prevalence of VG among young adolescents over time. Nevertheless, the similarity of the figures supports the sample’s representativeness and underscores the universality of VG use in our socio-cultural environment.

A higher proportion of males than females reported playing VG (96.6% compared to 51.2%), a more significant gap than in previous studies (e.g., 69.1% males compared to 50.4% females [22]), although this study employed a markedly distinct sample. The figures indicate that players switch between different types of games, with males using a greater variety than females (see Table 2). Action and Adventure games are the most frequently played, being the only types played by more than half of all participants (see Table 3). These are followed by Sports and MMO, with the others remaining below 40% in terms of use, and 8 of the 13 even being below 30%. These statistics contrast with those of German adolescents [21], who preferred, in this order, shooting games, single-player games, and MMOs. Notably, MMO games, considered in some studies [9] to be most highly related to IGD, are among the most frequently played VG in both that study and this one.

Table 4 displays the preferred game types, aligning with usage frequency but with greater variety in choices. Action and Adventure are the only ones favored by more than 50% of players; only four types are favored by at least 21% of participants. Generally, participants primarily play their favorite video games, along with some others.

Regarding VGP by gender, all types of games are played by a greater percentage of males than females, except for Simulators, Puzzle games, and Music and rhythm. Significant differences due to gender exist in 9 of the 13 types. The trend is clear, as in Action and Fighting games (Action, Sports, MMO, Strategy, Fighting, Shooting, and Role-Playing), the participation rates of males are at least triple those of females. In Simulators and Puzzle games, the participation rates of females are triple those of males. It is evident that males and females play different VG.

Although the number of days per week players play varies widely, the average indicates that they spend more days not playing than they spend playing. The average number of days suggests that they primarily play on weekends (Friday–Sunday), which would mean less interference with their academic work, although the study did not investigate this detail. Differences exist according to gender, with the average for males being one day longer than that for females. A positive relationship between age and playing days also appeared, r = 0.14.

Although it is impossible to establish the exact average number of hours they play, considering the hours are grouped into categories, this can be estimated at around 7.30 h per week (see Table 5). If this average is divided between the 3.30 days on which they say they play, we find moderate values, around 2.20 h a day, on the days they play. Although the averages are moderate, 55.4% say they play less than 6 h per week, 2.7% admit to playing more than 30 h a week, and 4% more than 25 h.

These averages change when considering gender, as the average (estimated) playing time for males is 9.36 h, compared to 4.03 for females. There are also significant differences in the percentage of males (11.6%) and females (1.2%) who play more than 16 h per week and more than 25 h per week (3.7% of males and 0.3% of females). In other words, the percentage of females who play VG is much lower, and those who play spend less time playing than males. The magnitudes indicated in this study seem lower than those in previous studies. Thus, Mérelle et al. [10] indicated an average of 15 h per week for males and 7 h for females, although their sample covers younger ages (12–15 years). Alternatively, Kiràly et al. [22] found that 33.8% play more than 3 h a day and 11.2% more than 7 h. Regarding the hours of VGP, Papay et al. [23] indicated that players at a high risk of developing IGD are mainly males who play for more than 5 h a day. The participants in our study, on average, play significantly fewer hours than those considered potentially problematic, raising the question of whether playing less than 25 h per week (in our study, 20 h) can be deemed “normal” or “non-problematic.” Age showed a positive relationship with the number of hours played, but it only accounted for 2% of the explained variance (r = 0.14, r^2^ = 0.020).

In terms of gaming devices, the preferred choice is video game consoles, followed by mobile phones (refer to Table 6). When considering gender, differences emerge; males lean toward playing on a video game console, while females show a preference for mobile phones. The use of mobile phones raises concerns due to their portability, enabling young adolescents to play anywhere. In contrast, video game consoles are a less conspicuous option outside their usual location. The increasing speed, reliability, and lower cost of internet connections from mobile phones suggest a growing preference for this device. However, it is foreseeable that traditional devices like video game consoles and computers may see a reduction in use. The minimal preference for TVs as gaming devices may be attributed to the limited penetration of smart TVs and their lack of portability.

The preferred gaming location is at home (87.6%), with other places receiving values below 5%, and no significant differences based on gender. This preference likely stems from the cost advantages and Wi-Fi speed, compared to the restrictive nature and cost of mobile data. It may also be influenced by the limited portability of video game consoles. Notably, participants prefer playing at home, and they tend to use mobile phones more than other devices like computers, tablets, or TVs, which offer larger and better screens. Mobility, potentially providing greater privacy, especially when playing in one’s room, appears more crucial than the quality of the display.

Regarding the social aspects of gaming, 44.1% express a preference for playing alone, while only 17.0% prefer company, either with one person (14.4%) or in a group (2.6%). In contrast, 39.0% prefer online company, either individually (15.4%) or in a group (23.6%). It is noteworthy that when participants prefer group play, it is predominantly online (23.6%) rather than in person (2.6%). These figures suggest that video game playing is fundamentally a solitary activity, and when played in company, the preference is for others not to be physically present. Gender differences are evident, with males favoring playing alone (36.7%) or in an online group (33.0%), while females lean toward playing alone (59.5%) or with someone physically present (22.0%). Notable is the high percentage of females preferring solo play compared to males and the substantial percentage of males preferring online group play compared to only 4.7% of females. When considering preferences for online gaming (individually or in a group), males account for 57.0%, compared to females at 14.4%. These findings indicate that the primary preference for males is to play accompanied online, while females prefer solo play. The majority of participants (87.2%) engage in online gaming (44.4% exclusively), and 55.6% play offline (12.8% exclusively). Online video game playing is likely to become increasingly dominant due to the ease and affordability of internet connections and the accessibility of popular games over the internet (e.g., Fortnite). These observations support the idea that differentiating between online and offline gaming may have limited value, as suggested by Porter et al. (2010), despite the DSM-5 (IGD) profiling focusing exclusively on online gaming. Once again, gender differences emerge, with males showing a preference for online play. Very few males prefer offline play compared to 26.4% of females. Finally, females express a preference for both modes (52.1%), compared to 38.3% of males.

Throughout the study, we underscore the importance of identifying distinct patterns among video game (VG) players in adolescents, a demographic group with a heightened susceptibility to developing addictive behaviors. These profiles offer valuable insights for constructing conceptual and empirical foundations that elucidate why some players encounter issues while others do not in their use of VG. This knowledge will facilitate the development of psychopathological models that can provide a more comprehensive understanding of VG-related problems. Enhancing precision in comprehending this phenomenon, particularly considering gender and maturation period differences, will enable more accurate identification of pertinent risk factors for each player. This, in turn, facilitates the establishment of predictive models and personalized interventions tailored to individual realities. Ultimately, these findings can serve as a foundation for future research endeavors investigating VG player profiles, the impact of VG on young people, the development of healthy and problematic behaviors, as well as interventions aimed at the prevention or resolution of issues associated with VG. Moreover, the empirical assessment of therapeutic programs addressing gambling problems in VG will furnish pertinent information for identifying underlying therapeutic processes. Additionally, we recommend the launching of prevention campaigns on VG and internet addiction and its impact on mental and physical health.

The results of this study, while consistent with the existing literature, are limited to the Spanish context. Within this context, the sample’s representativeness provides reasonable assurance for extrapolating the results. Another limitation of this study is the reliance on self-reporting data collected at schools. It would be beneficial to verify whether the reports from young adolescents correspond to their actual behavior. In this regard, gathering corroborating information from parents or guardians could be valuable in assessing the alignment of the data. However, the anonymous nature of the information, for ethical reasons, makes this process challenging.

## 5. Conclusions

In summary, approximately three-quarters of young adolescents engage in video gaming (VG), with the proportion of females participating being nearly half that of males. VG players, on average, play almost four different types of games. The most played and favored games tend to be in the action genre, particularly Action and Adventure, Sports, and MMOs, although males and females exhibit different preferences. They play for a little over three days per week, for more than seven hours weekly, with males playing more than females.

Regarding internet gaming disorder (IGD), a higher frequency of gaming was associated with higher IGDS9-SF scores for both men and women, although the relationship was more pronounced in women. Thus, gender is a relevant variable when characterizing video game players: being male was associated with a higher proportion of players, and they prefer action games, engaging more frequently. Furthermore, in line with the expectations, males scored higher for IGD.

Young adolescents prefer playing on a game console (males) or mobile phone (females), in the comfort of their own homes, often alone and online. Small but significant increases in IDGS9-SF scores were found based on the device used (PC and console for men, tablet for women), the mode of connection (online connection), the company (online with one person for men, online in a group for women), and the location (playing alone for men, playing at school). These profiles can serve as a guide for families and educators to identify potentially risky behaviors and IGD problems. Of course, before discussing risky behavior, it would be necessary to screen and evaluate each particular case. However, understanding the typical behavior of young adolescents is useful for initial observations.

## Figures and Tables

**Table 1 ijerph-20-07155-t001:** Educational level.

	N	%
1º ESO (equivalent to 7th grade, US system)	281	18.5
2º ESO (equivalent to 8th grade, US system)	421	27.8
3º ESO (equivalent to 9th grade, US system)	360	23.7
4º ESO (equivalent to 10th grade, US system)	224	14.8
1º y 2º Bachillerato (equivalent to 11th grade, US system)	170	11.2
FP Grado Básico (equivalent to Basic Professional Training, US system)	30	2.0
FP Grado Medio y Superior (equivalent to Advanced Professional Training, US system)	30	2.0
Total	1516	100

Note: 1º, 2º, and 3º ESO: 11 to 15 years (middle school or secondary education); 4º ESO, 1º Bachillerato: 15 to 17 years (high school or higher education).

**Table 2 ijerph-20-07155-t002:** Number of different types of video games played, by gender.

Gender	N	M	DT	Mdn	IRQ
Men	1025	3.90	2.45	3.00	3.00
Women	491	3.26	2.15	3.00	2.00
Total	1516	3.70	2.37	3.00	3.00

**Table 3 ijerph-20-07155-t003:** Percentage of use of each type of video game.

	Gender		
Type of Video Game	Men	Women	Total
	N (%)	N (%)	N (%)
Action and Adventure	738 (**48.7**)	237 (15.6)	975 (64.3)
Sports	573 (**37.8**)	151 (10.0)	724 (47.8)
Massively Multiplayer Online (MMO)	516 (**34.0**)	96 (6.3)	612 (40.4)
Shooters	532 (**35.1**)	47 (3.1)	579 (38.2)
Driving games	362 (23.9)	150 (9.9)	512 (33.8)
Platforms	249 (16.4)	170 (**11.2**)	419 (27.6)
Simulators	175 (11.5)	206 (**13.6**)	381 (25.1)
Fighting games	270 (**17.8**)	68 (4.5)	338 (22.3)
Puzzle games	101 (6.7)	229 (**15.1**)	330 (21.8)
Strategy games	190 (**12.5**)	47 (3.1)	237 (15.6)
Role games	159 (**10.5**)	37 (2.4)	196 (12.9)
Music and rhythm	61 (4.0)	119 (**7.8**)	180 (11.9)
Gambling games	75 (4.9)	45 (3.0)	120 (7.9)
Total	1025 (67.6)	491 (32.4)	1516 (100)

Note: N = 1516. Percentages based on the number of cases that answer “Yes” to each type of game. Note that this is a multiple-choice question. Answer percentages may add up to over 100%. In bold, the percentage of cases significantly higher than expected if gender and type of game were independent (results for chi-squared test and adjusted standardized residuals; the results are significant at the α level of 0.05).

**Table 4 ijerph-20-07155-t004:** Percentages of favorite game type.

	Gender		
	Men	Women	Total
Video Game Type	N (%)	N (%)	N (%)
Action and Adventure	664 (**43.8**)	222 (14.6)	886 (58.4)
Sports	534 (**35.2**)	117 (7.7)	651 (42.9)
Massively Multiplayer Online (MMO)	454 (**29.9**)	91 (6.0)	545 (35.9)
Shooters	499 (**32.9**)	34 (2.2)	533 (35.2)
Driving games	203 (13.4)	102 (6.7)	305 (20.1)
Platforms	126 (8.3)	131 (**8.6**)	257 (17.0)
Puzzle games	36 (2.4)	216 (**14.2**)	252 (16.6)
Simulators	64 (4.2)	185 (**12.2**)	249 (16.4)
Fighting games	112 (7.4)	41 (2.7)	153 (10.1)
Strategy games	115 (**7.6**)	34 (2.2)	149 (9.8)
Music and rhythm	21 (1.4)	91 (**6.0**)	112 (7.4)
Role games	87 (**5.7**)	21 (1.4)	108 (7.1)
Gambling games	44 (2.9)	45 (**3.0**)	89 (5.9)
Total	1025 (67.6)	491 (32.4)	1516 (100)

Note: N = 1516. Percentages based on the number of cases that choose a type of game as one of their three favorites. Note that this is a multiple-choice question. Answer percentages may add up to over 100%. In bold, the percentage of cases significantly higher than expected if gender and type of game were independent variables (results for chi-squared test and adjusted standardized residuals; the results are significant at the α level of 0.05).

**Table 5 ijerph-20-07155-t005:** Average hours of video game playing per week.

	Gender		
	Men	Women	Total
Hours	N (%)	N (%)	N (%)
More than 30 h	37 (**2.4**)	4 (0.3)	41 (2.7)
26–30	19 (**1.3**)	0 (0.0)	19 (1.3)
21–25	48 (**3.2**)	6 (0.4)	54 (3.6)
16–20	71 (**4.7**)	7 (0.5)	78 (5.1)
11–15	152 (**10.0**)	19 (1.3)	171 (11.3)
6–10	270 (**17.8**)	44 (2.9)	314 (20.7)
2–5	345 (22.8)	193 (**12.7**)	538 (35.5)
Less than 1 h	83 (5.5)	218 (**14.4**)	301 (19.9)
Total	1025 (67.6)	491 (32.4)	1516 (100)

Note: N = 1516. In bold, cells where the proportion of cases is greater than expected if the variables were independent (results for chi-squared test and adjusted standardized residuals; the results are significant at the α level of 0.05).

**Table 6 ijerph-20-07155-t006:** Preferences for playing video games, overall and by gender.

		Gender		
		Men	Women	Total
		N (%)	N (%)	N (%)
Device	Pc	182 (**12.0**)	67 (4.4)	249 (16.4)
	Video game console	467 (**30.8**)	79 (5.2)	546 (36.0)
	Mobile phone	200 (13.2)	251 (**16.6**)	451 (29.7)
	Tablet	67 (4.4)	50 (**3.3**)	117 (7.7)
	TV	109 (7.2)	44 (2.9)	153 (10.1)
Place	In your house	910 (**60.0**)	418 (27.6)	1328 (87.6)
	Friend’s house	38 (2.5)	22 (1.5)	60 (4.0)
	School	44 (2.9)	20 (1.3)	64 (4.2)
	Playhouses (recreation centers)	23 (1.5)	10 (0.7)	33 (2.2)
	On the street/in transit	10 (0.7)	21 (**1.4**)	31 (2.0)
Company	Alone	376 (24.8)	292 (**19.3**)	668 (44.1)
	One person (physically)	111 (7.3)	107 (**7.1**)	218 (14.4)
	One person (online)	182 (**12.0**)	51 (3.4)	233 (15.4)
	In group (physically)	18 (1.2)	21 (**1.4**)	39 (2.6)
	In group (online)	338 (**22.3**)	20 (1.3)	358 (23.6)
Connection mode	Online	568 (**37.5**)	105 (6.9)	673 (44.4)
	Offline	64 (4.2)	130 (**8.6**)	194 (12.8)
	Both online/offline	393 (25.9)	256 (**16.9**)	649 (42.8)
	Total	1025 (67.6)	491 (32.4)	1516 (100)

Note: N = 1516. In bold, cells where the proportion of cases is greater than expected if the variables were independent (results for chi-squared test and adjusted standardized residuals; the results are significant at the α level of 0.05).

**Table 7 ijerph-20-07155-t007:** IDGS9-SF score descriptive statistics, by video game playing preferences and gender.

	Gender		M	SD	df_1_, df_2_	F	*p*	η^2^
Device	Men	Pc	18.7	7.3	4, 1020	2.89	0.022	0.011
		Video game console	18.4	6.5				
		Mobile phone	17.1	5.6				
		Tablet	16.8	6.1				
		TV	17.6	6.2				
	Women	Pc	14.7	5.3	4, 486	2.9	0.022	0.023
		Video game console	13.5	4.7				
		Mobile phone	13.0	5.0				
		Tablet	15.1	6.8				
		TV	14.5	5.7				
Place	Men	In your house	18.0	6.2	4, 1020	4.66	0.001	0.018
		Friend’s house	17.3	7.3				
		School	15.8	6.5				
		Playhouses (recreation centers)	20.7	9.1				
		On the street/in transit	24.2	14.6				
	Women	In your house	13.7	5.4	4, 486	2.13	0.076	0.017
		Friend’s house	14.2	4.3				
		School	15.7	5.9				
		Playhouses (recreation centers)	14.1	5.0				
		On the street/in transit	11.0	2.3				
Company	Men	Alone	17.9	6.6	4, 1020	5.54	<0.001	0.021
		One person (physically)	15.9	5.1				
		One person (online)	19.5	7.3				
		In group (physically)	17.5	5.0				
		In group (online)	18.1	6.1				
	Women	Alone	13.8	5.4	4, 486	8.36	<0.001	0.064
		One person (physically)	13.4	4.6				
		One person (online)	11.2	3.3				
		In group (physically)	13.5	3.4				
		In group (online)	19.0	7.5				
Connection mode	Men	Online	18.7	7.0	2, 1022	1.03	<0.001	0.019
		Offline	15.3	5.8				
		Both online/offline	17.5	5.6				
	Women	Online	14.6	5.4	2, 488	3.29	0.038	0.013
		Offline	12.8	4.4				
		Both online/offline	13.7	5.6				

Note: N = 1516.

## Data Availability

We provide information on how the sample size was established, disclose any data exclusions, and detail all study measurements. We describe all statistical analyses and the software used, and the data and syntax can be obtained from the authors upon a reasonable request. The research materials can be accessed at osf.io/nrv45. It is important to note that the study and data analyses were not preregistered, although the study protocol was submitted to the University’s Ethics Committee.

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
