# Peer review of "Video Game Playing and Internet Gaming Disorder: A Profile of Young Adolescents"

_ijerph, 2023, doi:10.3390/ijerph20247155_

Round 1
Reviewer 1 Report
Comments and Suggestions for Authors
The purpose of this paper is to profile the video game playing behaviour of children and adolescents from 12 to 16 years of age attending school in Madrid. Furthermore, authors explored the relationship between some characteristics of this profile and IGD. This paper needs some revision and improvements before being accepted for the publication.
· Authors should provide a structured abstract for more clarity such as background, purpose, methodology, recommendation and conclusion.
· Please provide full forms of all abbreviations like for DSM.
· Provide definition of IGF in first paragraph.
· The authors should mention the purpose of the study into past tense
· I would suggest full form of children and adolescent through out the paper, using so many acronyms confuse the reader
· Remove full form of IGD from line 125
· Please mention if ethics approval was obtained?
· Did you take consent or accent from participants and their parents?
· Mention VGP instead of full form in line 234, check this through out your paper.
· Provide some recommendations from your study.
· Authors need to revise the whole paper to make it grammatically correct and lessen the amount of abbreviations used through out the paper.
Comments on the Quality of English LanguagePlease review paper throughout.
Author Response
Reviewer #1
The purpose of this paper is to profile the video game playing behaviour of children and adolescents from 12 to 16 years of age attending school in Madrid. Furthermore, authors explored the relationship between some characteristics of this profile and IGD. This paper needs some revision and improvements before being accepted for the publication.
Thank you for your dedication in reviewing this manuscript. We will try to address all the issues and suggestions from the three reviewers to improve the quality of the paper.
- Authors should provide a structured abstract for more clarity such as background, purpose, methodology, recommendation and conclusion.
We have proceeded to change the abstract based on the recommendations you provided earlier.
- Please provide full forms of all abbreviations like for DSM.
We have now included the full form of DSM (Diagnostic and Statistical Manual of Mental Disorders). We will check the text for abbreviated forms that have not been properly introduced.
- Provide definition of IGF in first paragraph.
We have provided the definition of IGD that appears in the diagnostic manual DSM-5. We added, in the first paragraph: ‘The Diagnostic and Statistical Manual of Mental Disorders, DSM-5 [4], defined internet gaming disorder (IGD) as “a pattern of excessive and prolonged Internet gaming that results in a cluster of cognitive and behavioral symptoms, including progressive loss of control over gaming, tolerance, and withdrawal symptoms, analogous to the symptoms of substance use disorders”.’
- The authors should mention the purpose of the study into past tense
We have corrected that, leaving only the acronym.
- I would suggest full form of children and adolescent through out the paper, using so many acronyms confuse the reader
We have reverted all the "C&A" acronyms to "children and adolescents."
- Remove full form of IGD from line 125
We have corrected that.
- Please mention if ethics approval was obtained?
The approval was already stated in the "Institutional Review Board Statement" mandatory section. We have now added the name of the university.
- Did you take consent or accent from participants and their parents?
In the Procedure section, it says, "Finally, Gamertest was taken by the pupils from the 37 selected schools who obtained their guardians’ consent."
- Mention VGP instead of full form in line 234, check this through out your paper.
The mention in line 234 is for video game players instead of playing, hence the full form. The tables have also the full form to make them self-explanatory.
- Provide some recommendations from your study.
As suggested, we have proceeded to incorporate practical implications and future directions for research, based on our findings
- Authors need to revise the whole paper to make it grammatically correct and lessen the amount of abbreviations used through out the paper.
After addressing the reviewers' concerns and suggestions, we have thoroughly reviewed the manuscript to improve readability and language quality.
Reviewer 2 Report
Comments and Suggestions for Authors
I thank the authors for the work carried out, which appears to be precise and methodologically strong. The authors involved a large representative sample of children and adolescents and conducted a study that suggests interesting results regarding video game playing and internet disorder.
I think the article is complete and I have few recommendations to suggest.
1. I suggest an English language review with a native speaker.
2. I ask the authors whether their work was conducted during, before or after the pandemic and whether the pandemic context may have influenced the results.
3. I would like to read at least one paragraph on the relationship between amount of time spent on the internet and psychological disorders in children. Is it the time spent online or other aspects that are a risk factor? Furthermore, I would like to read at least one paragraph that is a summary on the relationship between the internet and the psychological needs of children (https://doi.org/10.1016/j.addbeh.2021.107204).
4. I suggest the authors dedicate a separate paragraph (with its own subtitle) dedicated to practical implications and future directions for research. In particular, I believe that the authors can extend the practical implications of their research and argue them more thoroughly.
.
Author Response
Reviewer #2
I thank the authors for the work carried out, which appears to be precise and methodologically strong. The authors involved a large representative sample of children and adolescents and conducted a study that suggests interesting results regarding video game playing and internet disorder.
I think the article is complete and I have few recommendations to suggest.
Thank you for your kind words; we are pleased that this topic has sparked your interest. Also, thank you for your dedication in reviewing this manuscript. We will try to address all the issues and suggestions from the three reviewers to improve the quality of the article.
- I suggest an English language review with a native speaker.
We have reviewed the grammar and style of the text after addressing the reviewers' questions. We hope that this revised version allows for a smoother and more appropriate reading experience.
- I ask the authors whether their work was conducted during, before or after the pandemic and whether the pandemic context may have influenced the results.
The data collection for the sample took place before the Covid-19 pandemic; therefore, in this case, it has no impact on the results obtained.
- I would like to read at least one paragraph on the relationship between the amount of time spent on the internet and psychological disorders in children. Is it the time spent online or other aspects that are a risk factor? Furthermore, I would like to read at least one paragraph that is a summary on the relationship between the internet and the psychological needs of children (https://doi.org/10.1016/j.addbeh.2021.107204).
As proposed, we have incorporated these two ideas into the text.
- I suggest the authors dedicate a separate paragraph (with its own subtitle) dedicated to practical implications and future directions for research. In particular, I believe that the authors can extend the practical implications of their research and argue them more thoroughly.
We considered it a very good suggestion and have proceeded to incorporate an entire paragraph into our text.
Reviewer 3 Report
Comments and Suggestions for Authors
Brief summary : This interesting study focuses on the new diagnosis introduced in the DSM-5, known as Internet Gaming Disorder. Usage profiles of video game players (1516 adolescents aged between 12 and 16 years old in the city of Madrid, Spain) were established. The authors identified that male tended to play more frequently than females and that higher gaming frequency is associated with higher Internet Gaming Disorder scores, especially in females. Console gaming was more frequently associated amongst male were as mobile gaming was more frequently identified in females.
Please see my comments below.
Introduction :
- The introductory sentence is miss-leading considering the authors mentin that there has been little concern about addictions in children and adolescents. There is a vast body of literature about addictions in the youth, especially drug usage.
- Furthermore, video game playing exists since the middle of the 1950s. Qualifying it as new technologies (line 31) is therefore incorrect.
- The link between the knowledge gap and the main objective is currently unclear. Why does the lack of a videogame profile in children and adolescent justifies a study in the 12 to 16 years old population (young adolescents only)?
- While the authors attempted to provide the readership with several aspects of IGD in the introduction, the current structure is very hard to follow. Re-structuring the introduction as the following is encouraged :
o 1st paragraph : What is Internet Gaming Disorder?
o 2nd paragraph : Defining the current literature on IGD in the youth population.
o 3rd paragraph : Outlining the main problem (knowledge gap) and why it is important to answer this gap.
o 4th paragraph : Main objectives and hypothesis
Materials and Methods :
- Participants : details on the inclusion & exclusion criteria, choice of population, recruitement methodology is missing.
- Instruments : Validation & Reliability (metrics on the used instruments – for GamerTest) must be presented. Otherwise there is no guarantee that the results could be reproduced or that the instruments adequatly capture the items being assessed.
- Choice of instruments and rationale behind their use is missing. Why these two tests adequately capture elements of a player’s profile?
- Table 1 is a result. Typically, participants characteristics and demographics are presented in the results section.
- Data analysis : line 160 : why is N for non-players calculated if they are not part of the analysis?
- What are the demographics information being collected? There might be several differences between playing habits of 12 years old respondents as compared to 16 years old respondents.
Results :
- Participants demographics should be presented. A table for mis encouraged.
- The presented data is unclear because the elements provided by the instruments were not presented in the methodology. Presenting the main categories of the instruments in the methodology could ease the readership in understanding the main reported results.
- Statistical significancy must presented in the Tables for the reported data.
Discussion :
- It is unclear for many of the discussion points the hypothesis or main arguments which may support why the presented results differ in some cases from the literature. An exemple are lines 289-292. The authors are encouraged to provide such explanations.
- In the limitations of the study, the authors state that these results cannot be generalized outside the Spanish context. At this point, it is still unclear why the targetted population are 12-16 years old adolescents whereas the question of interest focuses on children and adolescent? This could be further explicited.
Minor comments :
- Many sentences are incomprehensible. For example, line 39-40 have several typos and semantic issues. Major revision of the grammatical aspects of the manuscript should be conducted.
- Many different reference styles are used across the manuscrip (i.e : line 85 uses APA, where as line 92 uses Vancouver). Please follow MDPI guidelines.
- Many common nouns are capitalized and introduced as major terms such as Chidren and Adolescents (line 30), Video Game Player (line 32) or Video Games (line 33). Since these are not titles or entities as such and are common nouns, there should not be capitalization.
- Ethics : the ethics declaration states the following : Ethics Committee of the University's Faculty of Psychology. Which University? This should be part of the declaration.
Comments on the Quality of English LanguageMany sematic and grammatical issues.
Author Response
Reviewer #3
Brief summary : This interesting study focuses on the new diagnosis introduced in the DSM-5, known as Internet Gaming Disorder. Usage profiles of video game players (1516 adolescents aged between 12 and 16 years old in the city of Madrid, Spain) were established. The authors identified that male tended to play more frequently than females and that higher gaming frequency is associated with higher Internet Gaming Disorder scores, especially in females. Console gaming was more frequently associated amongst male were as mobile gaming was more frequently identified in females.
Please see my comments below.
Thank you for your thorough dedication in reviewing this manuscript. We will try to address all the issues and suggestions from the three reviewers to improve the quality of the paper.
Introduction :
- The introductory sentence is miss-leading considering the authors mentin that there has been little concern about addictions in children and adolescents. There is a vast body of literature about addictions in the youth, especially drug usage.
Fully in agreement with your assessment, we have proceeded to change this fact.
- Furthermore, video game playing exists since the middle of the 1950s. Qualifying it as new technologies (line 31) is therefore incorrect.
Once again, in full agreement with the assessment, we have also proceeded to change it appropriately.
- The link between the knowledge gap and the main objective is currently unclear. Why does the lack of a videogame profile in children and adolescent justifies a study in the 12 to 16 years old population (young adolescents only)?
We have chosen the age range of 12 to 16 years as it represents a more homogeneous age group and encompasses the entire period of “enseñanza secundaria obligatoria” (compulsory secondary education) in Spain, equivalent to grades 7th through 10th in the United States.
Following your review, we have proceeded to change the term used previously, "children and adolescents," to "young adolescents," as outlined by UNICEF in its categorization by developmental periods.
- While the authors attempted to provide the readership with several aspects of IGD in the introduction, the current structure is very hard to follow. Re-structuring the introduction as the following is encouraged :
o 1st paragraph : What is Internet Gaming Disorder?
The reviewer raises a fundamental point. While we discussed the criteria that allow us to delineate the construct, we overlooked the simplest and most evident step, which is to provide a formal definition. Now we have provided the definition of IGD that appears in the diagnostic manual DSM-5. We added, in the first paragraph: ‘The Diagnostic and Statistical Manual of Mental Disorders, DSM-5 [4], defined internet gaming disorder (IGD) as “a pattern of excessive and prolonged Internet gaming that results in a cluster of cognitive and behavioral symptoms, including progressive loss of control over gaming, tolerance, and withdrawal symptoms, analogous to the symptoms of substance use disorders”.’
2nd paragraph : Defining the current literature on IGD in the youth population
The suggested changes have been implemented to enhance coherence in the text. Due to the comprehensive nature of the text, we kindly refer you directly to the document for the review.
o 3rd paragraph : Outlining the main problem (knowledge gap) and why it is important to answer this gap.
The suggested changes have been implemented to enhance coherence in the text. Due to the comprehensive nature of the text, we kindly refer you directly to the document for the review.
o 4th paragraph : Main objectives and hypothesis
The suggested changes have been implemented to enhance coherence in the text. Due to the comprehensive nature of the text, we kindly refer you directly to the document for the review.
Materials and Methods :
- Participants : details on the inclusion & exclusion criteria, choice of population, recruitement methodology is missing.
The sample was selected using stratified random sampling at schools in the city of Madrid. Data on the student population for the city’s 21 districts—their age, school year, and type of schooling (public school, private school, and state-subsidized school)—were retrieved from the website of the city government’s statistics service. Then a stratified representative sample was derived matching the distribution of students by districts, type of school, and school year from 12 to 16 years old. Five independent evaluators with psychology degrees were trained to correctly administer the assessments through Gamertest. Schools were divided into groups by district and type of school and randomly ordered. Then, for each district and type of school, the first school on the list was contacted through a detailed letter, with a follow-up call soon thereafter, and asked to provide access to the entire set of classes required for the district. If the school refused, the next school on the list was contacted. The time of year the request was made, towards the end of the school year, was a common reason schools refused to collaborate. Once a school agreed to participate in the study, the evaluators delivered informed consent forms for the children’s parents/guardians, and a date was set for the evaluator to visit the school to perform the assessment in the classes that had been chosen using the stratified random sampling. After collecting the informed consent forms from the parents/guardians, the assessments were administered in groups on computers in each school’s computer room, allowing approximately 30–40 min for students to complete them. The participants’ responses were anonymously collected and coded directly in the computerized database.
- Instruments : Validation & Reliability (metrics on the used instruments – for GamerTest) must be presented. Otherwise there is no guarantee that the results could be reproduced or that the instruments adequatly capture the items being assessed.
The reviewer is correct that, as currently written, it seems like information is missing. For this paper, the instrument that requires a report on its psychometric qualities is the IGDS9-SF, and this is reflected in its corresponding paragraph. The Gamer Test is not a single instrument; it is a platform where participants can respond to questions from various instruments. While data were collected with other instruments that also require the evaluation of their reliability and validity (engagement, attitudes, cognitions, self-control/impulsiveness, and emotional unease), these instruments are not used here, and therefore, their qualities are not described. What we used, however, are the sociodemographic and gaming habits data. We have clarified this in the new version of the manuscript, adding “For this study, we have only used items (1), (2), and (3); the psychometric properties of the measurement instrument are detailed in the following section”.
- Choice of instruments and rationale behind their use is missing. Why these two tests adequately capture elements of a player’s profile?
As we mentioned in the previous response, the Gamer Test is an online platform that includes various questionnaires and measurement instruments, among which we used the questionnaire on habits and demographic characteristics, as well as the IGD9-SF. Since our intention was to profile the video game user and relate these characteristics to gaming problems as defined in the DSM-5 manual, we found the tools used to be appropriate.
- Table 1 is a result. Typically, participants characteristics and demographics are presented in the results section.
It's not uncommon for them to be presented in the Methods section to provide context when explaining the Procedure. On this occasion, we have preferred this structure. This way, Table 1 does not unnecessarily enlarge the Results section, which goes directly to the characterization of gaming habits and, if applicable, their relationship with gambling problems.
- Data analysis : line 160 : why is N for non-players calculated if they are not part of the analysis?
The reviewer is absolutely right. Initially, some differences were calculated between players and non-players, and from there, power analysis was conducted. However, ultimately, this study was conceived as a descriptive study regarding habits and characteristics, and inferential in terms of comparing groups of players. We have conducted a new power analysis, focusing only on players, and for the comparison among six groups (the statistically more conservative analysis among the possible options in the study). The revised sentence for the analysis is: "A statistical power of 1 - β = .869 was calculated in a one-way ANOVA test, comparing players (n = 1,516) among five groups (as in the 'preferred place for video gaming' comparison), setting α at .050, even assuming a small effect size f = .10."
- What are the demographics information being collected? There might be several differences between playing habits of 12 years old respondents as compared to 16 years old respondents.
As mentioned earlier, responses were collected through the Gametest application. In this application, various sociodemographic variables are inquired about (gender, age, school year, name of the school, employment status and marital status). However, for the purposes of this study, we specifically focused on gender, age, and school year
Results :
- Participants demographics should be presented. A table for mis encouraged.
As we have mentioned, this time demographic characteristics regarding gender, age, and educational cycles are in the Participants section, leaving the Results section to discuss habits and the relationship of all characteristics with IGD. However, if the reviewer insists that we move that information from Participants to Results, we will do so.
- The presented data is unclear because the elements provided by the instruments were not presented in the methodology. Presenting the main categories of the instruments in the methodology could ease the readership in understanding the main reported results.
We believe that this issue is clarified after specifying which instruments we used from those offered by the Gamer Test. The reliability statistics (split-half reliability) and internal consistency (alpha and omega) of the IGDS9-SF, as well as its unifactorial structure, are detailed in the Instruments and Variables section. Additionally, it is an instrument whose detailed psychometric properties are published (Sánchez-Iglesias et al., 2020). In any case, we present in the manuscript the IGDS9-SF scoring scale with percentiles, so that the reader can get an idea of how large the difference in averages is between the groups defined by demographic and gaming habit variables.
- Statistical significancy must presented in the Tables for the reported data.
Some of the presented results are purely descriptive (Tables 1 and 2). Table 7 already includes the significance level of the conducted comparisons. The remaining tables contain comparisons of proportions. Although categories with a statistically higher or lower proportion than expected are marked in bold, it is true that the significance level is not specified. This was done for greater clarity in reading. However, since tables should be self-explanatory, we have specified the significance level at which results were declared statistically significant.
Discussion :
- It is unclear for many of the discussion points the hypothesis or main arguments which may support why the presented results differ in some cases from the literature. An exemple are lines 289-292. The authors are encouraged to provide such explanations.
The study we are comparing with was conducted in 2014. Being our most recent study, the availability of video games has increased in these intervening years, leading to a greater disparity in their selection. More games are now available, and they are more easily accessible. On the other hand, their sample consists exclusively of 16-year-old individuals who are Hungarian (considering cultural differences). We have proceeded to explain it briefly in the text.
- In the limitations of the study, the authors state that these results cannot be generalized outside the Spanish context. At this point, it is still unclear why the targetted population are 12-16 years old adolescents whereas the question of interest focuses on children and adolescent? This could be further explicited.
Indeed, the results are difficult to extrapolate to other countries due to the idiosyncrasies of education in each territory, influenced by factors such as language, culture, and specific educational policies promoted by governments. While this statement may seem evident, we have preferred to make it explicit in the manuscript. About the age range, we have already discussed the reason previously.
Minor comments :
- Many sentences are incomprehensible. For example, line 39-40 have several typos and semantic issues. Major revision of the grammatical aspects of the manuscript should be conducted.
We will thoroughly revise the quality of the grammatical and typographical aspects of the text.
- Many different reference styles are used across the manuscrip (i.e : line 85 uses APA, where as line 92 uses Vancouver). Please follow MDPI guidelines.
By default, we drafted the initial manuscript in APA format and later adapted it to the MDPI format. Although we use a reference manager, some elements from the previous format may remain. Once we address the queries from all three reviewers, we will thoroughly review the entire text to avoid these formatting issues. Additionally, we will likely seek assistance from the journal's editors and their style review team.
- Many common nouns are capitalized and introduced as major terms such as Chidren and Adolescents (line 30), Video Game Player (line 32) or Video Games (line 33). Since these are not titles or entities as such and are common nouns, there should not be capitalization.
This is another aspect of formatting and language that we will review once the reviewers' queries are addressed. Thanks to your feedback, we hope that the new version of the manuscript will be much easier to read.
- Ethics : the ethics declaration states the following : Ethics Committee of the University's Faculty of Psychology. Which University? This should be part of the declaration.
In the initial version, we drafted a manuscript with the intention of maintaining anonymity for peer review. Now, we have added the missing information "...Ethics Committee of the Complutense University of Madrid's Faculty of Psychology...".
Round 2
Reviewer 3 Report
Comments and Suggestions for Authors
The authors responded adequately to my comments and suggestions. I have no further suggestions considering the manuscript significantly improved.
Comments on the Quality of English LanguageNil